# Genetic Diversity of Durum Wheat (*Triticum turgidum* L. ssp. *durum*, Desf) Germplasm as Revealed by Morphological and SSR Markers

**DOI:** 10.3390/genes14061155

**Published:** 2023-05-26

**Authors:** Temesgen Dagnaw, Behailu Mulugeta, Teklehaimanot Haileselassie, Mulatu Geleta, Rodomiro Ortiz, Kassahun Tesfaye

**Affiliations:** 1Department of Microbial, Cellular and Molecular Biology, Addis Ababa University, Addis Ababa P.O. Box 1176, Ethiopia; temesgen.dagnaw@aau.edu.et; 2Institute of Biotechnology, Addis Ababa University, Addis Ababa P.O. Box 1176, Ethiopia; behailu.mulugeta@slu.se (B.M.); teklehaimanot.hselassie@aau.edu.et (T.H.); 3Department of Plant Breeding, Swedish University of Agricultural Sciences, P.O. Box 190, SE-23422 Lomma, Sweden; mulatu.geleta.dida@slu.se (M.G.); rodomiro.ortiz@slu.se (R.O.); 4Ethiopian Bio and Emerging Technology Institute, Addis Ababa P.O. Box 5954, Ethiopia

**Keywords:** durum wheat, genetic diversity, landraces, morphological traits, SSR markers

## Abstract

Ethiopia is considered a center of origin and diversity for durum wheat and is endowed with many diverse landraces. This research aimed to estimate the extent and pattern of genetic diversity in Ethiopian durum wheat germplasm. Thus, 104 durum wheat genotypes representing thirteen populations, three regions, and four altitudinal classes were investigated for their genetic diversity, using 10 grain quality- and grain yield-related phenotypic traits and 14 simple sequence repeat (SSR) makers. The analysis of the phenotypic traits revealed a high mean Shannon diversity index (H′ = 0.78) among the genotypes and indicated a high level of phenotypic variation. The principal component analysis (PCA) classified the genotypes into three groups. The SSR markers showed a high mean value of polymorphic information content (PIC = 0.50) and gene diversity (h = 0.56), and a moderate number of alleles per locus (Na = 4). Analysis of molecular variance (AMOVA) revealed a high level of variation within populations, regions, and altitudinal classes, accounting for 88%, 97%, and 97% of the total variation, respectively. Pairwise genetic differentiation and Nei’s genetic distance analyses identified that the cultivars are distinct from the landrace populations. The distance-based (Discriminant Analysis of Principal Component (DAPC) and Minimum Spanning Network (MSN)) and model-based population stratification (STRUCTURE) methods of clustering grouped the genotypes into two clusters. Both the phenotypic data-based PCA and the molecular data-based DAPC and MSN analyses defined distinct groupings of cultivars and landraces. The phenotypic and molecular diversity analyses highlighted the high genetic variation in the Ethiopian durum wheat gene pool. The investigated SSRs showed significant associations with one or more target phenotypic traits. The markers identify landraces with high grain yield and quality traits. This study highlights the usefulness of Ethiopian landraces for cultivar development, contributing to food security in the region and beyond.

## 1. Introduction

Durum wheat (*Triticum turgidum* L. ssp. *durum*, Desf.) is an amphidiploid (AABB) domesticated species (2*n* = 4*x* = 28), originated through intergeneric hybridization and polyploidization between *Triticum urartu* (A genome) and *Aegilops speltoides*-related species (B genome) [1]. The primary origin of durum wheat is thought to have occurred between 12,000 and 10,000 years ago in the West Levantine region [2]. However, Pecetti et al. [3] stated the distinctiveness of the Ethiopian durum wheat germplasm, with a unique morphology and with no sign of allelic similarity to the primary origin in the Levantine. In addition, Ethiopian farmers developed durum wheat anew through the further domestication of emmer wheat, which gave rise to *T. turgidum* ssp. *aethiopicum* [4].

Ethiopia is one of the main centers of durum wheat genetic diversity [5,6,7,8,9]. However, landraces have been neglected for over five decades, and their production has declined from 60% of total production in the 1970s to 10–15% in 2018. This has led to a significant reduction in its genetic diversity [10]. The decline in the production of the landraces is due to farmers’ preference for the cultivation of improved durum wheat cultivars, bread wheat cultivars, and teff [11,12,13]. In addition, Ethiopian durum wheat landraces have been rarely included in the breeding programs seeking to develop improved varieties. For instance, according to an established pedigree of cultivars released in Ethiopia from 1900 to 2012 by the Wheat Atlas Organization [14], less than 2% of the improved cultivars comprised genes from Ethiopian durum wheat landraces, while more than 98% of the cultivars were foreign materials. The lack of an adequate characterization of durum wheat landraces for economically relevant traits might have prevented their wider use in breeding programs. 

The success of breeding programs relies on significant genetic variation in source populations [15]. Higher genetic variation levels for the target characteristic(s) within a species imply increased opportunities to select superior genotypes from a population [16] or to have more novel traits and alleles resilient to unpredictable climate changes and new end-user demands [17]. It is vital to explore genetic variability in landrace populations in order to understand the genetic factors behind their adaptation and to identify beneficial alleles [18]. Therefore, determining the genetic diversity of Ethiopian durum wheat landraces can provide highly valuable information that will help us to broaden the genetic base of the germplasm used in breeding programs.

Morphological markers are useful indicators of genetic diversity, but they are not the most reliable, as they are highly affected by the environment [19]. In contrast, DNA markers can pinpoint multiple genes involved in regulating complex traits without being influenced by the environment. Hence, combining marker-assisted selection (MAS) with other crossbreeding methods can enhance the overall gain per given time, as well as the precision and efficiency of breeding programs [15]. Molecular markers such as AFLP, RAPD, ISSR, and SSR have been used to assess diversity across several crops [20,21,22]. Simple sequence repeat (SSR) markers, also called microsatellites, are preferred markers compared to many other types of DNA markers for genetic diversity analyses of crops because of their high polymorphism [23], co-dominance, high rate of transferability across closely related species [24], amenability to simple PCR-based assays [25], abundance, and high reproducibility [26]. 

The present study assessed the genetic diversity of Ethiopian durum wheat based on quality- and yield-related morphological traits and SSRs previously reported to be associated with these traits. Ethiopian durum wheat genetic diversity has been previously assessed using SSR markers [5,6,27,28,29]. However, most of these investigations were conducted without a clear distinction between durum wheat and other tetraploid wheat crops, considering the geographical origin of the germplasm in the country [5,27,29], and sufficiently representing the major durum wheat-producing areas [28]. The study by Asmamaw et al. [6] was inclusive in terms of area coverage, but the markers were not associated with quality- and yield-related traits. Moreover, the correlation between genetic diversity assessed by SSR markers and morphological markers has not been investigated for Ethiopian durum wheat. Therefore, this study evaluated the Ethiopian durum wheat germplasm in terms of (1) genetic variation and population structure, using 14 SSR loci, and (2) the extent and pattern of morphological diversity using 10 quality- and yield-related traits. 

This study provided valuable information on the genetic structure and extent of the genetic diversity of Ethiopian durum wheat genotypes. In addition, few landraces showed good levels of performance in yield and quality traits, which highlights the importance of landraces for developing cultivars. This will serve as a basis for further research aimed at identifying superior genotypes for the improvement of desirable traits, such as end-use quality.

## 2. Materials and Methods

### 2.1. Plant Material and Field Experiment

In this study, 104 genotypes of Ethiopian durum wheat were used, which included 94 landraces and 10 cultivars selected from those used in the study by Dagnaw et al. [9] on the basis of their grain quality and yield. The landraces representing different geographical regions were obtained from the Ethiopian Biodiversity Institute (EBI) (Figure 1) and characterized for two consecutive years (2017–2018) at Sinana Agricultural Research Center (SARC) to maintain their trueness to type for further characterization. The modern cultivars were obtained from Debrezeit Agricultural Research Center (DzARC) and SARC (Appendix A). We grouped the genotypes into thirteen populations, three regions, and four altitudinal classes depending on their geographical zone, their Regional State, and the altitude ranges of their collection sites in Ethiopia, respectively. The field experiment was conducted at Chefe-Donsa (08°44′ N and 39°09′ E, 2450 m.a.s.l) and Sinana (07°07′ N and 40°10′ E, 2400 m.a.s.l) research centers during the 2019/2020 cropping season using alpha lattice design with two replications. 

### 2.2. Measurement of Morphological and Grain Quality Traits

The data on spikelets per spike (SPS, count), plant height (PLH, in cm), spike density (SPD), and vitreosity (VTR) were recorded following the descriptor list for wheat [30]. The data on days to heading (DH, in days) and days to maturity (DM, in days) traits were taken following the description provided by Hailu et al. [31]. The thousand-kernel weight (TKW, in g) was recorded by weighing 250 kernels and multiplying by four. The grain yield (GY, in tha^−1^) was recorded by weighing total economic seed harvested from a plot and converted to hectare. The GY and TKW traits were weighed at the standardized moisture content of 12%. The grain moisture content (GMC, %), gluten content (GC, %) and grain protein content (GPC, %) were measured using near-infrared spectroscopy (NIRS) as described in AOAC [32] method 997.06 (32.2.03 A).

### 2.3. SSR-Based Molecular Characterization

#### 2.3.1. Genomic DNA Extraction

Five healthy seeds from each of the 104 genotypes were randomly selected and planted in pots in a greenhouse at Addis Ababa University (AAU), Ethiopia. Two-week-old leaf samples were collected and dried using silica gel. Genomic DNA was extracted using the cetyltriethyl ammonium bromide (CTAB) method following a protocol optimized by Abdie et al. [33] at the Plant Genetics Laboratory, AAU. DNA quality and quantity were checked using Nano-drop 2000 and 1% agarose gel electrophoresis.

#### 2.3.2. SSR Marker Selection

Fourteen SSR markers associated with grain quality- and yield-related traits of durum wheat from previous research [34,35,36,37,38,39,40] were used in this study (Table 1).

#### 2.3.3. PCR Amplification

The PCR amplification was carried out in a PCR tube with 10 μL total reaction volume containing 7.3 μL nuclease-free water, 1 μL 10× PCR buffer with 17.5 mM MgCl_2_, 0.25 μL dNTPs (2.5 mM), 0.15 μL of each forward and reverse primer (10 mM), 0.15 μL Taq-polymerase enzyme (3 U/μL) and 1 μL of DNA template (100 μL/ng). The PCR condition was adjusted to 4 min preheating and initial denaturing at 94 °C; 35 cycles of 30 s denaturation at 94 °C, 30 s annealing at a temperature specific to each primer pair, and 90 s primer extension at 72 °C; and 8 min final primer extension at 72 °C. 

#### 2.3.4. Gel-Electrophoresis and Data Scoring

The PCR products were separated on 8% polyacrylamide gel socked in 0.5× TBE buffer for 3.5 h at a constant voltage (110 V) and current (30 mA). The electrophoresed gels were stained by socking and gentle shaking in 0.5 μg/mL EtBr solution for 20 min and rinsed with distilled water two times. The gels were finally visualized using the Bio-rad Gel-Doc EZ gel documentation machine (Bio-Rad Laboratories, Hercules, CA, USA), and the molecular weight of the bands on the gels was estimated in base pairs, using Image-Lab 6.1 software (Bio-Rad Laboratories, Hercules, CA, USA). 

### 2.4. Statistical Analysis

#### 2.4.1. Morphological Data Analysis

The frequencies of phenotypic classes of each morphological trait were calculated based on all genotypes as a single group, populations, regions, and altitudinal classes (Appendix A). Using the frequency data, the standardized Shannon diversity index (H′) was calculated for traits, populations, regions, and altitudinal classes as described in Eticha et al. [41]:$$H\prime =\left(\sum_{i=1}^{R}pi\left(lnpi\right)\right)/ln(n)$$
where n is the number of phenotypic classes of the trait and pi is the proportion of ith phenotypic class. H′ was estimated for all qualitative traits and all genotypes.

Principal component analysis (PCA) was computed to identify the diversity pattern among genotypes and the contribution of morphological traits. The data of morphological traits were scaled to unit variance and means of zero using the scale function of Stats package [42] in R and analyzed for PCA using prcomp of Stats package [42] in R. The scatterplot of individual genotypes and morphological traits was plotted using the fviz_pca_biplot function of the factoextra package [43] in R. To determine the relationship between morphological and geographical distance, a Mantel test was carried out based on Pearson’s product-moment correlation using the mantel function of the Vegan package [44] in R.

#### 2.4.2. SSR Data Analysis

To estimate the resolution of the 14 SSR markers used in the present study, a genotype accumulation curve was plotted using the genotype_curve function of the pegas package [45] in R. Mean diversity indices of the loci and populations and analysis of molecular variance (AMOVA) were computed using GenAlEx version 6.5 software [46]. Major allele frequency (MAF), gene diversity (h), and polymorphic information content (PIC) were computed using Power marker version 3.25 software [47]. The pairwise Nie’s genetic distance and genetic differentiation between populations were analyzed and plotted using hierfstat [48] and ggplot2 [49] packages in R, respectively. 

To determine genetic relationships between genotypes, the Minimum Spanning Network (MSN) based on Bruvo’s distance [50] was computed using poppr [51] and adegenet [52] packages in R. The Discriminant Analysis of Principal Component (DAPC) was carried out using the adegenet [52] package in R. The relationship between SSR-based genetic distance and geographical distance was determined by a Mantel test based on Pearson’s product-moment correlation using the mantel function of Vegan package [44] in R.

The population structure was analyzed using STRUCTURE version 2.3 software [53]. To estimate the optimum value of K, a burn-in period was set to 100,000, followed by 200,000 Markov Chain Monte Carlo (MCMC) replications for K from 1 to 10 with 20 iterations for each K. The optimal K-value was determined as described in Evanno et al. [54] using Structure Harvester Web v0.6.94 [55]. The packaging and plotting of the optimum K population structure inferences were carried out using the beta version of CLUMPAK software [56]. 

## 3. Results

### 3.1. Morphological Diversity

#### 3.1.1. Shannon Diversity Index

The standardized Shannon diversity index (H′) of the 10 morphological traits indicated significant phenotypic variation with an overall mean of 0.78. The value of H′ is classified as high (H′ ≥ 0.60), intermediate (0.40 ≤ H′< 0.60), and low (H′ < 0.40), according to Eticha et al. [41]. All phenotypic traits were highly polymorphic (H′ ≥ 0.60) except DM (H′ = 0.40) and VTR (H′ = 0.45) (Table 2). The frequency distribution of the traits’ phenotypic classes showed the dominance of genotypes with late maturity, low gluten content, and high vitreousness. For the remaining traits, the phenotypic classes were fairly proportional (Appendix A). The studied genotypes were grouped into thirteen populations, three regions, and four altitudinal classes based on their geography of origin in Ethiopia. With respect to the populations, high H′ values were recorded for all traits except for DM (H′ = 0.40) and VTR (H′ = 0.45) (Table 2). Among the traits, DM had the lowest H′ values across most populations. West Gojam, East Harerge, and Bale populations did not show variation for vitreousness (H′ = 0.00), while the Central Tigray (H′ = 0.80) and Southern Tigray (H′ = 0.73) populations showed high variation for this trait. When the Shannon diversity index was compared across populations, H′ values were high (H′ ≥ 0.60) for all populations, with a mean value of 0.71. The H′ for GL and GPC in all populations was high, with the highest value recorded for the West Shewa population. The North Shewa population was highly diverse in DH, TKW, and DM. The frequency of different phenotypic classes of traits differed across the populations (Appendix A). The Arsi population was characterized by genotypes with a late heading, low SPS, low GY, very high GL, high GPC, lax spike, and high vitreousness. The West Shewa population had genotypes with extended DM, high SPS, long PLH, low TKW, and low GY at high frequency. The cultivar population exhibited genotypes with an early heading, intermediate SPS, short PLH, very high TKW, highest GY, low GL, low GPC, and a dense spike at a higher frequency than other populations. 

The estimates of the within-region diversity index were high (H′ > 0.60) for all traits except DM (H′ = 0.39) and VTR (H′ = 0.50) (Table 2). The Oromia region has shown the highest diversity index compared to the other regions for most of the investigated traits. The regions showed a similar distribution of phenotypic classes of traits, except for the highest frequency of longer PLH, and high TKW, GC, and GPC, in the Oromia region (Appendix A). Morphological traits also varied from one altitudinal class to another. Genotypes originating from the lowest altitude (<2000 m.a.s.l.) showed higher frequency for late DH, low SPS, high TKW, low GY, and low GPC than other altitudinal classes, whereas genotypes collected from the highest altitude (>2800 m.a.s.l.) showed higher frequency for high SPS, low TKW, high GPC and dense spike than other altitudinal classes. All traits were highly polymorphic within each altitudinal class, with H′ values of above 0.70, except DM and VTR. The corresponding values for these two traits were less than 0.50 (Table 2). Genotypes originating from low altitudes (<2000 m.a.s.l) scored the highest diversity index for DM, SPS, TKW, and GL, while PLH, GPC, and SPD showed the highest diversity index in genotypes collected from high altitudes (>2800 m.a.s.l). The highest diversity index values for DH and VTR were observed within the altitudinal class of 2001–2400 m.a.s.l., and GY was observed within the altitudinal class of 2401—2800 m.a.s.l.

#### 3.1.2. Principal Component Analysis (PCA)

Principal component analysis (PCA) was used to understand the contributions of 10 agronomic and quality traits in terms of magnitude and direction to the variation and grouping of the 104 genotypes. The first two principal components (PCs) explained 49.4% of the total phenotypic variation among the genotypes (Figure 2). The first principal component (PC1), which explained 32.8% of the total variation, was associated positively with TKW, GY, GC, and GPC, and negatively with the remaining traits. In addition, the second principal component (PC2) accounted for 16.6% of the total variation and was associated positively with GPC, GC, DH, DM, PLH, and VTR, and negatively with the remaining traits. Although most landraces resided close to the bi-plot origin, five landraces (two from Arsi (75 and 79), two from East Shewa (46 and 47), and one from West Shewa (56)) were separated for their highest contribution to GC and GPC. All cultivars (except 99) and three landraces (28, 89, and 90) were distinctly separated from the landraces for their highest contribution to TKW and GY. The Mantel test identified a significant correlation between morphology-based genetic distance and geographical distance (RXY = 0.34, *p* = 0.001) across durum wheat genotypes.

### 3.2. SSR Diversity

#### 3.2.1. Level of Polymorphism of the SSR Loci

The 14 SSR markers used in this study were distributed across the first six chromosomes of the A genome and the seven chromosomes of the B genome of durum wheat (Table 1). As shown by the genotype accumulation curve, the 14 SSRs together are able to reveal the maximum levels of differentiation among the genotypes included in our sampling (Appendix A). In addition, the polymorphic information content (PIC), which describes the extent of information retrieved from each locus, ranged from 0.17 (Xwmc256) to 0.70 (Xgwm294), with a mean value of 0.50 (Table 3). All SSR markers were highly polymorphic and showed a total of 56 alleles, which ranged from 2 (Xwmc256 and Xgwm513) to 6 (Xbarc240), with an average of 4.0 alleles per locus. The gene diversity (h) ranged from 0.19 (Xwmc256) to 0.74 (Xwms294), with a mean value of 0.56 (Table 3). The highest MAF was recorded for locus Xwmc256, which had the lowest number of alleles, Shannon diversity index (I), h, and PIC. In contrast, locus Xgwm294 had the lowest MAF and the highest h and PIC. 

#### 3.2.2. Genetic Diversity across Populations

The percentage of polymorphic loci (PPL) ranged from 86% (Southern Tigray) to 100% (North Gondar, East Shewa, West Shewa and Arsi), with an average value of 95% (Table 4). The number of alleles per locus (Na) in the populations ranged from 2.29 (West Gojam) to 3.57 (Cultivars), with a mean value of 2.73. The effective number of alleles (Ne) ranged from 1.79 (Central Tigray) to 2.68 (Cultivars), with a mean value of 2.06. The Shannon diversity index (I) values ranged from 0.62 (Central Tigray) to 1.04 (Cultivars), with a mean value of 0.77. The observed heterozygosity (Ho) values ranged from 0.15 (East Harerge) to 0.37 (Cultivars). In addition, none of the populations had private alleles.

#### 3.2.3. Genetic Distance and Differentiation between Populations

The genetic distance and differentiation between the populations were estimated by pairwise population measures of Nei’s genetic distance and Fst coefficient. The highest pairwise Nei’s genetic distance was obtained between Cultivars and South Wello (0.53), and Cultivars and Bale (0.51) (Figure 3). The pairwise Nie’s genetic distance value of Cultivars, East Shewa, South Tigray, and Arsi populations was relatively high compared to other populations, while it was lower for Central Tigray, Bale, East Gojam, and North Gondar populations. The genetic differentiation (Fst) coefficients obtained for most pairwise comparisons of populations (Figure 3) were above the threshold of significant differentiation among populations (Fst = 0.12; *p* < 0.001). The highest genetic differentiation was for Cultivars versus Central Tigray, Bale, and South Wello. In general, the Cultivars population was the most differentiated from the other populations, which is also seen in Sothern Tigray, East Shewa and South Wello populations. However, North Gondar, Central Tigray, East Gojam, and Bale showed lower pairwise genetic differentiation than other populations. 

#### 3.2.4. Analysis of Molecular Variance (AMOVA)

AMOVA revealed 12%, 3%, and 3% of total variation among populations, regions, and altitudinal classes, respectively. In contrast, 88%, 97% and 97% of the total genetic variations were observed within populations, regions, and altitudinal classes, respectively, with an overall significant (*p* < 0.001) genetic differentiation coefficient (Fst) and high gene flow between populations (Nm = 1.83), regions (Nm = 8.37) and altitudinal classes (Nm = 7.70) (Table 5). In addition, the SSR-based genetic distance of genotypes was not significantly correlated with their corresponding geographical distance (RXY = 0.005, *p* = 0.412).

#### 3.2.5. Minimum Spanning Network (MSN) and Discriminant Analysis of Principal Component (DAPC)

The MSN was used to reveal the genetic relatedness between the 104 Ethiopian durum wheat genotypes representing 13 populations using Bruvo’s distance method. The MSN analysis grouped the genotypes into two clusters at 0.1 cut point distance (Figure 4). Most landraces were categorized in the first cluster (C1). However, the second cluster (C2) contained all cultivars (except 99), and six landraces (1, 13, 41, 43, 76, and 90). The DAPC differentiated the populations into two groups. One was positioned at the center, comprising all landrace populations, while the second was placed to the left, comprising only Cultivars (Figure 5). The first two dimensions, dimension one (Dim1) (30.6%) and dimension two (Dim2) (18.9%), explained 49.5% of the total phenotypic variation among genotypes. The biplot of DAPC showed that the placement of the populations followed the geographical locations of constituting genotypes, except that the North Shewa and Arsi populations were separated from populations in close geographical proximity. 

#### 3.2.6. Population Structure

The Bayesian model-based analysis of population structure across 104 genotypes identified two groups as an appropriate number of clusters based on the Evanno method [54] (Figure 6). The STRUCTURE bar graph, which provides information on the level of admixture in the studied samples, reveals admixture in all genotypes and populations, although to different extents. Admixture levels are high for East Shewa, West Shewa, Arsi, and Southern Tigray populations. 

#### 3.2.7. Relationship between Markers and Associated Traits

All SSR markers showed significant relationships with their associated traits. For instance, the Xgwm46 locus significantly correlates with GPC and GY (Figure 7). Genotypes with the allele combination of 152/152 showed the highest GPC and the lowest GY, while 168/168 showed the highest GY and moderate GPC. The Xwmc256 locus showed significant relationships with GY and DH. Genotypes with the allele combination of 117/117 had the highest GY and the lowest DH. The Xgwm493 locus showed a significant relationship with GPC, DH, PLH, and GY. At this locus, genotypes with allelic combinations of 168/145 and 179/145 showed the lowest DH and PLH, and the highest GY. 

## 4. Discussion

### 4.1. Magnitudes of SSR Markers-Based Diversity

This study estimated the extent of genetic diversity and the population structure of 104 Ethiopian durum wheat genotypes using SSR and morphological markers. The SSRs were used to evaluate genetic diversity and genotype population structure. It is extremely valuable to determine genetic variation at loci contributing to grain quality and yield in durum wheat to identify novel alleles that can be used for breeding purposes to improve the crop. A total of 14 SSR markers distributed across the durum wheat genome were used to estimate the genetic diversity and population structure of the durum wheat genotypes. Several previous reports have shown that SSRs are among the most effective DNA-based markers for genetic diversity studies and the fingerprinting of crop plants [23,25,26,57]. The SSR markers used in this study were highly polymorphic. The observed mean Na (4.0) coupled with the high mean value of PIC (0.50) and gene diversity (0.56) (Table 3) signify the usefulness and informativeness of the SSR markers used in the present study to scan the genetic diversity of durum wheat. At the Xbarc240, Xgwm294, Xgwm47, Xbarc12, and Xgwm371 loci, five to six alleles were detected, resulting in higher I, h, and PIC values than those recorded at other loci (Table 3). This indicates a high efficiency of these five loci in differentiating different durum wheat genotypes. 

Compared to this study, a higher number of alleles were observed by Mondini et al. [28] (Na = 8.7) across 23 Ethiopian durum wheat landraces using 28 SSR markers (Table 3). Alamerew et al. [27] also reported a mean Na of 7.9 for 22 SSR markers utilized in 12 durum wheat accessions. Furthermore, a mean Na of 6 was reported by Asmamaw et al. [6] in a study involving 160 durum wheat accessions and 12 SSR markers. The fewer alleles observed in this study than previous reports might have arisen due to differences in markers and genotypes. For this study, genotypes were selected using morphological trait-based indicators of durum quality traits. Genotypes ranking highly in quality traits were selected. This potentially increased the similarity between the genotypes, leading to lower diversity among them. Moreover, marker selection was also based on their association with grain quality traits reported in previous studies. Therefore, both cases increased the similarity between the genotypes and resulted in lower differences in the number of observed alleles per locus compared to previous studies. Generally, the levels of genetic diversity in durum wheat reported in these studies were greatly influenced by the selected SSR loci, the total number of markers used, and the type and geographical distribution of genotypes. Overall, the multiple alleles identified at several loci in this study confirmed the high genetic diversity in Ethiopian durum wheat.

Several studies carried out on durum wheat from other countries also reported different mean values for Na, depending on the number of SSR markers and the landraces or cultivars used. For example, Christov et al. [58] reported a mean observed Na of 6.9 for 32 SSR markers across 90 Bulgarian durum wheat accessions. Marzario et al. [59] found a mean Na of 4.1 for 44 SSR markers across 164 Italian durum wheat genotypes. Ouaja et al. [60] estimated a mean Na of 9.9 for 10 SSR markers among 304 Tunisian durum wheat landraces. These studies on Ethiopian and non-Ethiopian durum wheat germplasm suggest durum wheat’s capacity to adapt to different climatic and agroecological conditions worldwide.

### 4.2. Genetic and Morphological Diversity within Populations

The number and frequency of alleles at a given locus in a given population provide valuable information about the genetic diversity of that particular population. The observed high mean values of most diversity indices (Na = 2.73; Ne = 2.06; I = 0.77; PPL = 95%) across the populations reveal high genetic diversity within Ethiopian durum wheat populations. Previously, high genetic polymorphism in Ethiopian durum wheat populations was reported by Mondini et al. [28] (Na = 4.87; PPL = 91%) across 9 populations and by Asmamaw et al. [6] (Na = 5.9; I = 1.39; PPL = 88%) across 15 populations. Our findings and previous research as cited above reflect the existence of genetically diverse populations in the Ethiopian durum wheat gene pool. Cultivars scored the highest values among the populations for all genetic diversity indices, thereby indicating the cultivars are genetically distinct and were developed based on genotypes with diverse genetic backgrounds. This suggests that the cultivars could have been developed through several crosses between diverse durum wheat individuals from different sources that may have led to multiple allelic combinations [61]. Among landrace populations, the East Shewa population scored the highest values for most diversity indices, which indicates the population holds higher genetic diversity than other landrace populations. Similarly, a high genetic diversity of durum wheat populations using SSR markers was previously noted in various parts of the world. For instance, Marzario et al. [59] reported Na = 3.09 and He = 0.53 in Italian durum wheat genotypes, and Ouaja et al. [60] estimated Ne = 1.89, I = 0.62 and PPL = 75.8% in Tunisian durum wheat landraces.

Besides molecular diversity, this study further explored the morphological diversity of the genotypes based on 10 quality- and yield-related traits, and identified high genetic diversity. The levels of morphological diversity obtained in this study were higher than those reported by Mengistu et al. [62] and Hailu et al. [31] in the Ethiopian durum wheat collections. A similar level of morphological diversity was also observed in 304 Tunisian durum wheat landraces (H′ = 0.80) using 12 morphological traits [60]. Hence, the high level of morphological diversity revealed in the Ethiopian durum wheat collection suggests a potential resource to be exploited in durum wheat breeding programs. The highest diversity observed in the West Shewa population for GC and GPC suggests the significance of the germplasm from that part of the country in relation to improving quality traits in durum wheat. The highest H′ value identified in the Oromia region for most traits indicates the region’s potential for encouraging durum wheat diversity in Ethiopia, which was also reported by Mengistu et al. [62]. The highest H′ values obtained from the lowest (<2000 m.a.s.l.) and the highest (>2800 m.a.s.l.) altitudinal classes also suggest the presence of high genetic diversity in these altitudinal classes, which could have come about due to several adaptation strategies of the crop to the respective agroecology. In previous research, interestingly, Eticha et al. [41] estimated high diversity at high altitudes (>2501 m.a.s.l.), and Mengistu et al. [62] noticed high diversity at low altitudes (1600–2000 m.a.s.l.), both of which were confirmed in the present study.

### 4.3. Genetic Differentiation across Populations

AMOVA revealed highly significant genetic variation among the studied populations, accounting for 12% of the total genetic variance (12.53) (Table 5), which is comparable with previous reports on durum wheat populations in Ethiopia by Mondini et al. [28] (18.76%), in Tunisia by Ouaja et al. [60] (19%), in the Mediterranean region by Amallah et al. [39] (16%), and in India by Arora et al. [63] (21.3%). The identified lower proportion of genetic variation among populations than within populations could be explained by the high gene flow between populations, which might have occurred due to the exchange of durum wheat germplasm between farmers and through market systems. The high genetic variation within populations observed in this study is mainly attributable to gene flow, mutation, and genetic recombination. Hence, desirable genotypes can be found within a given population for use as parents for crossbreeding. The estimated high gene flow among populations also led to lower genetic differentiation and genetic distance between some populations in this study, in agreement with that of Ouaja et al. [60]. The separation of cultivars from landraces based on morphological markers in this study and previous reports [62] was confirmed by SSR-based diversity analysis in this study. Thus, selecting parental genotypes from highly differentiated populations for crossbreeding might be more effective when seeking to improve durum wheat than using genotypes from less differentiated populations.

### 4.4. Genetic Relationship, Population Structure and Association with Phenotypic Traits

The MSN and DAPC analyses were used to identify the genetic relationship between the genotypes used in this study. In the MSN analysis, the landraces were divided into two clusters based on their geographical proximity, and with a distinct distinction between cultivars (Figure 4). Both phenotypic and SSR-based investigations grouped the cultivar genotype 99 with landraces, which could be due to the fact that the cultivar may share similar morphological traits and genetic backgrounds with landraces, or might have been mistakenly registered as a cultivar by curators. In addition, the grouping of some landraces with cultivars in both investigations may be due to landraces sharing similar morphological traits and genetic backgrounds with cultivars, or those genotypes might have been mistakenly registered as landraces by curators. Similarly, in a study by Slim et al. [64], 41 Tunisian landraces and 13 varieties were grouped into two clusters, in which varieties were clustered together with landraces. Arora et al. [63] also reported similar results, in which durum varieties with landraces were clustered together. According to Quinn and Keough [65], cluster-based analysis is more informative for closely related individuals, while principal component analysis is more sensitive to genetic distances among groups. The result of DAPC identified two groups in which the grouping followed the geographical proximities of the landraces (Figure 5). This indicates that genetically related or similar genotypes might have been cultivated in close geographical locations, which was confirmed by the high gene flow identified among populations, and the significant correlation between morphological trait-based genetic distance and geographical distance. The separation of cultivars from landrace populations suggests the uniqueness of cultivars that arise due to the accumulation of specific alleles in the cultivars through various generations of breeding and selection [61], and the underutilization of local landraces in breeding programs. The reason for the underutilization of landraces in breeding programs could be, among others, their tall height and late maturity [66]. However, the local landraces with high genetic diversity can serve as potential sources of new alleles for crop improvement [18]. The selection of landraces or parent lines with the best allelic combinations for local adaptation can also be facilitated by interacting with farmers’ traditional knowledge [67].

The PCA divided the genotypes into three groups based on the magnitude and direction of the contribution of different traits to the principal components (Figure 2). The positive contribution of GC and GPC, and the negative contribution of SPD and SPS to both principal components (PC1 and PC2), suggest the negative correlations of GC and GPC with SPD and SPS in durum wheat, as noticed by Dagnaw et al. [9]. Similar grouping patterns of morphological traits in a PCA biplot were also observed in previous research [68,69,70]. The separation of five landraces from other landraces in terms of their most significant contribution to GC and GPC suggests the high potential of these genotypes for use in improving the crop’s GC and GPC through crossbreeding. In addition, all cultivars except genotype 99 and three landraces (28, 89, and 90), which were distinctly separated from other landraces, were characterized by their high TGW and GY (Figure 2). Therefore, involving these cultivars and landraces in breeding programs would improve TKW and GY in durum wheat. Interestingly, the separation of all cultivars from landraces except genotype 99, and the grouping of genotype 90 (a landrace) with cultivars, were also observed in the SSR-based (DAPC and MSN) analysis. This may highlight the capacity of morphological markers used for durum wheat diversity analysis.

The model-based clustering method (STRUCTURE) classified the genotypes into two genetic groups with different admixture levels. Some landrace populations showed low admixture, but all received alleles from the two genetic groups (Figure 6). Although the studied genotypes showed similar morphological appearances in terms of the vitreous kernel, amber kernel color, high gluten content, high grain protein content, and moderate thousand-grain weight and grain yield, the populations were structured into two genetic groups. This grouping could be due to the relatedness of the genotypes for the traits associated with the SSR markers, the sharing of similar genetic backgrounds, and/or the high gene flow found among populations in this study (Nm = 1.83). Hence, the grouping based on genetic differences among genotypes at the investigated loci is attributable to mutation, recombination, and related genetic phenomena. Similarly, Christov et al. [58] across 90 Bulgarian durum wheat accessions and Kehel et al. [71] across Moroccan (98) and Syrian (90) durum wheat landraces found two genetic groups, along with some admixtures. In contrast, Arora et al. [63] reported more structured (K = 7) and genetically highly admixtured subpopulations for 319 Indian durum wheat varieties. Marzario et al. [59] also found six groups with few admixtures among 136 landraces and 28 varieties of Italian durum wheat accessions. Recently, Negisho et al. [8] analyzed 285 durum wheat genotypes comprising 215 landraces, 10 cultivars, and 10 advanced lines from Ethiopia, and 50 durum wheat lines from CIMMYT, using SNP markers, and grouped the whole genotypes into two population clusters representing mainly the landraces on the one hand, and cultivars, advanced Ethiopian and CIMMYT lines on the other. The differences in the population stratification of durum wheat collections in several countries suggest differences in the diversity of the gene pool sampled for the studies.

### 4.5. Relationship between SSR Markers and Associated Traits

The investigated SSR loci showed significant associations with one or more target phenotypic traits. The SSR Xgwm46 showed a significant association with GPC and GY (Figure 7), which are highly correlated traits [9]. This SSR was previously reported to be associated with GPC [39] and GY [36]. Among the genotypes bearing different allele combinations, selecting genotypes that are homozygous for the 152 bp allele (genotype 152/152) and that show the highest mean GPC would be beneficial to increasing GPC through crossbreeding, although their GY is the lowest. On the other hand, a homozygous genotype for the 168 bp allele (genotype 168/168) is suitable for use as a parent in crossbreeding for GY improvement with no significant negative effect on GPC. The Xwmc256 locus showed a significant relationship with GY and DH. The observed high grain yield and early heading in genotypes homozygous for the 117 bp allele (genotype 117/117) compared to other allelic combinations at the Xwmc256 locus imply they would be beneficial for use in durum wheat breeding for grain yield improvement. The locus Xgwm493 was previously reported to be associated with GPC [37], and DH, PLH, and GY [36], which are essential traits in durum wheat. Among the identified allelic combinations at this locus, 168/145 and 179/145 showed the lowest DH and PLH, and the highest GY. Hence, genotypes having these allelic combinations will donate alleles that contribute to developing early-heading shorter plants with high grain yields.

## 5. Conclusions

The molecular and phenotypic diversity analyses carried out in this study revealed considerable genetic variation in Ethiopian durum wheat genotypes. In both molecular and phenotypic diversity analyses, cultivars and landraces were distinguished clearly. The PIC value suggests the potency of the markers in detecting polymorphism. A total of 56 alleles were identified among studied genotypes. The SSR markers used were highly polymorphic, and revealed genetic variation within the populations and genetic differentiation between the populations. The lower observed heterozygosity compared to the expected heterozygosity at all loci is consistent with the inbreeding characteristics of durum wheat. The population structure analysis grouped the genotypes into two clusters. The grouping of landrace genotype 90 with cultivars in both morphological- and SSR-based investigations should be further evaluated and included in breeding programs that may facilitate the development of cultivars with better local adaptation. Interestingly, all SSR loci showed significant associations with one or more phenotypic traits, confirming previously reported associations. As a result, the favorable alleles at these SSRs can serve as markers for marker-assisted selection in durum wheat breeding programs, both for identifying genotypes that can serve as parents for crossbreeding and for pyramiding desirable characteristics of agronomic and quality traits into a standard breeding line.

## Figures and Tables

**Figure 1 genes-14-01155-f001:**
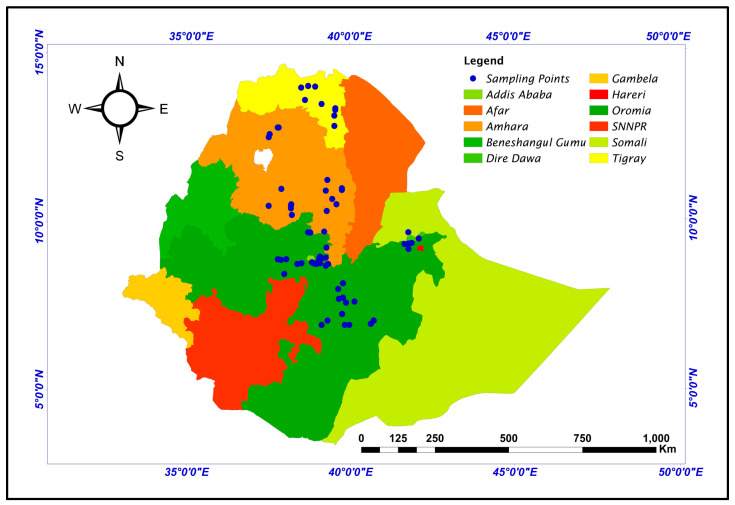
Map of Ethiopia showing the sample collection sites of landraces. The colored boxes represent the regions found in Ethiopia and the blue dots show sample collection sites.

**Figure 2 genes-14-01155-f002:**
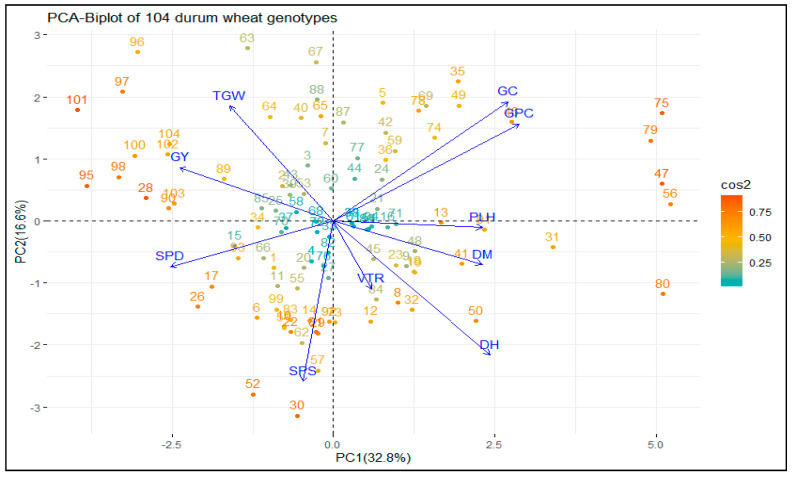
A biplot of principal component analysis (PCA) of 104 durum wheat landraces and cultivars with their contribution value based on 10 agronomic and quality traits. The genotypes are colored based on their contribution to the two principal components from green (0%) to red (100%). The length of the arrows is equivalent to the variance of the variables, whereas the angles between them (cosine) are equivalent to their correlations. DH is days to heading; DM is days to maturity; SPS is number of spikelet per spike; PLH is plant height; TKW is thousand-kernel weight; GY is grain yield; GC is gluten content and GPC is grain protein content. PC1 and PC2 are principal component one and two, respectively.

**Figure 3 genes-14-01155-f003:**
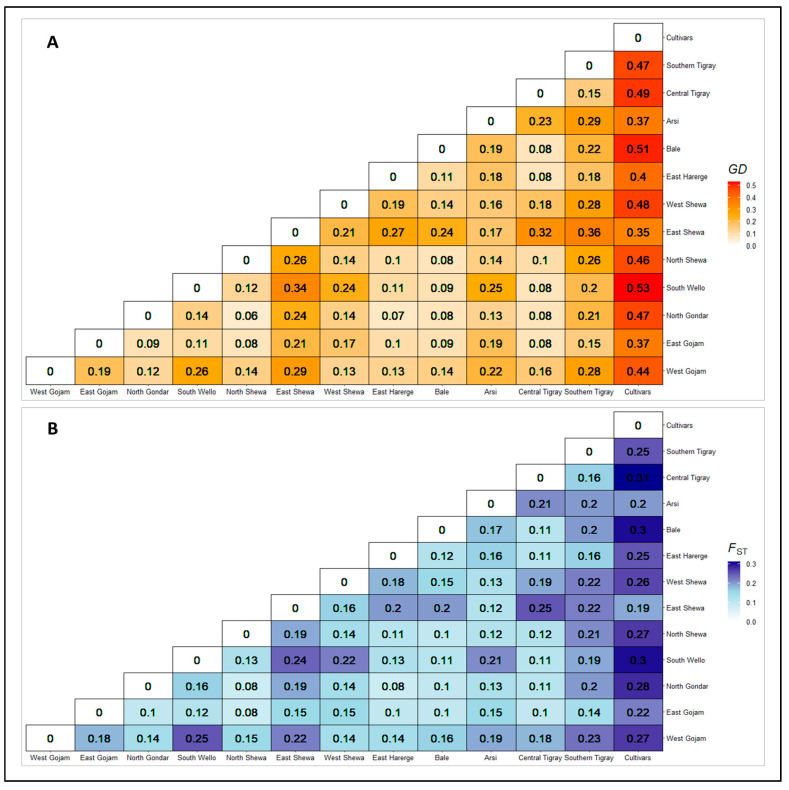
Pairwise Nie’s standard genetic distance (GD) (**A**) and genetic differentiation (F_ST_) (**B**) among studied populations. Darker colors indicate that populations are genetically distant, whereas lighter colors indicate that populations are genetically close.

**Figure 4 genes-14-01155-f004:**
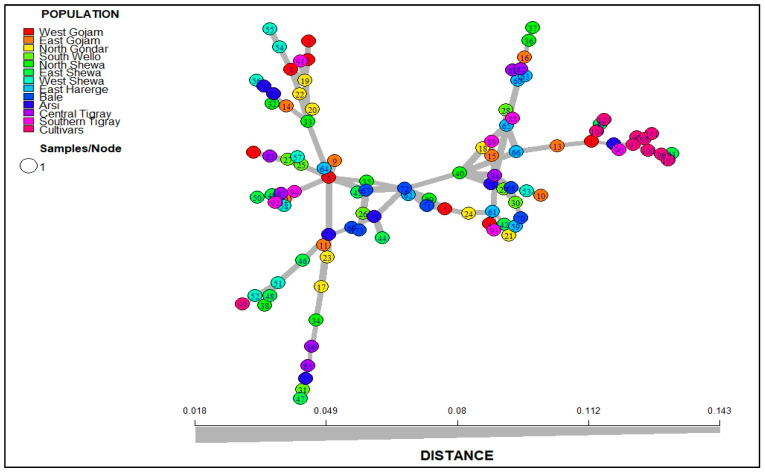
Minimum Spanning Network (MSN) constructed for the 104 Ethiopian durum wheat genotypes based on Bruvo’s distance estimated using 14 SSR markers. Each node represents a single genotype. The nodes are colored according to the populations. The thickness of the lines represents the degree of relatedness between genotypes.

**Figure 5 genes-14-01155-f005:**
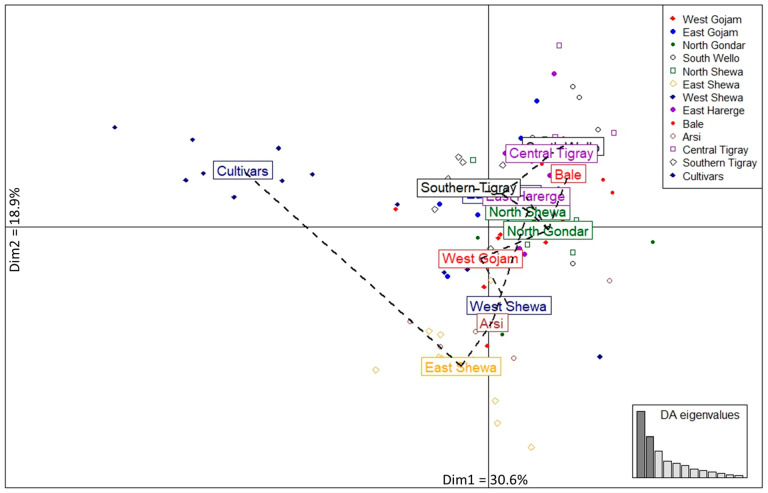
Biplot of discriminant analysis of principal component (DAPC) for thirteen studied populations. Different shapes and colors represent the populations. The bar plot at the right bottom corner shows the eigenvalues of identified dimensions. Dim1 and Dim2 are dimension one and two, respectively. DA is discriminant analysis.

**Figure 6 genes-14-01155-f006:**
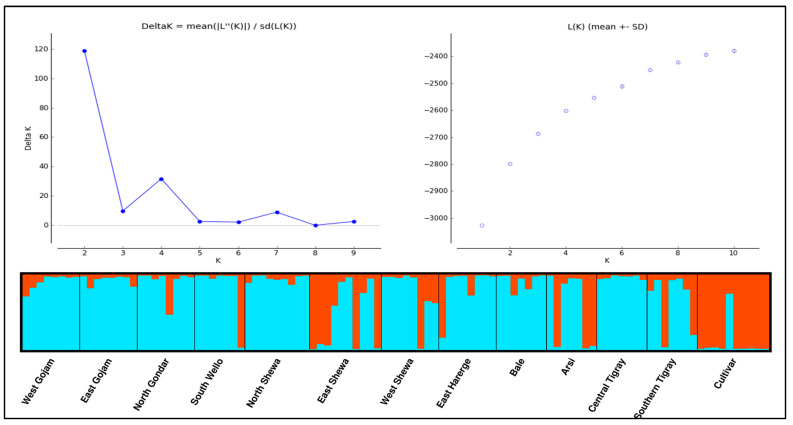
Population structure of 104 Ethiopian durum wheat genotypes. Biplots showing the optimal number of clusters (K) at two (**top left**) and Log likelihood versus the number of K (**top right**) based on Evanno et al.’s (2005) method, and a structure bar graph of the populations at K = 2, where the green and orange colors represent the two genetic groups (**bottom**).

**Figure 7 genes-14-01155-f007:**
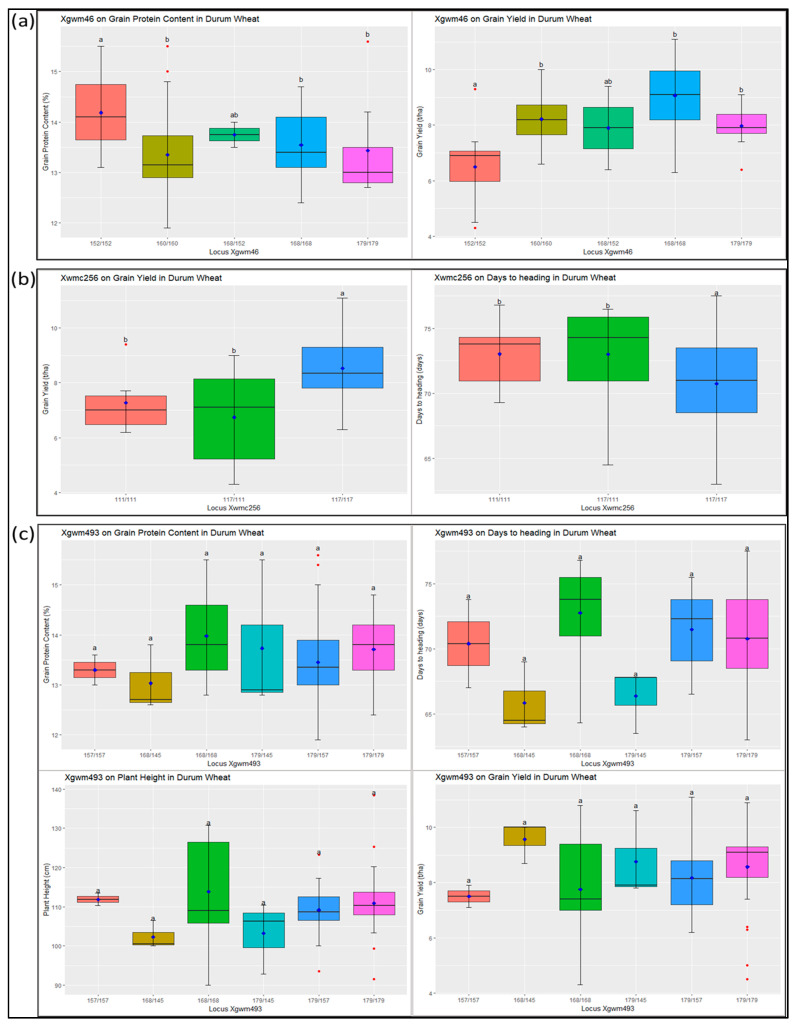
Boxplots depicting the association between different allelic combinations of (**a**) Xgwm46, (**b**) Xwmc256, and (**c**) Xgwm493 SSR loci with variation in agronomic and quality traits that were previously shown to have significant associations. The small rhombus within each boxplot shows the mean value while the small red circles outside the boxes represent outliers of corresponding traits. The letters ‘a’, ‘ab’ and ‘b’ indicate statistically significant differences at 0.01 level. ‘a’ has a mean that is statistically different from ‘b’; and ‘ab’ has a mean that is not statistically different from either ‘a’ or ‘b’.

**Table 1 genes-14-01155-t001:** List of simple sequence repeat (SSR) markers used in this study and their associated traits, as previously reported.

SSR	Chr. Position (cM)	Ta	Primer Sequence (5′-3′)		Associated Trait	References
Xwmc24	1AS (35.9)	52	Fw	GTGAGCAATTTTGATTATACTG	TKW	[34]
			Rv	TACCCTGATGCTGTAATATGTG	TW, GY	[36]
Xbarc240	1BL (46.4)	60	Fw	AGAGGACGCTGAGAACTTTAGAGAA	TW, YP	[34]
			Rv	GCGATCTTTGTAATGCATGGTGAAC		
Xgwm294	2AL (118.3)	61	Fw	GGATTGGAGTTAAGAGAGAACCG	DH, PLH, TKW	[36]
			Rv	GCAGAGTGATCAATGCCAGA		
Xgwm47.1	2BL (116.8)	61	Fw	TTGCTACCATGCATGACCAT	PLH, TKW	[36]
			Rv	TTCACCTCGATTGAGGTCCT		
Xbarc12	3AS (12.8)	56	Fw	CGACAGAGTGATCACCCAAATATAA	TKW	[36]
			Rv	CATCGGTCTAATTGTCAATGTA		
Xgwm493	3BS (10.7)	60	Fw	TTCCCATAACTAAAACCGCG	DH, PLH, TW, GY	[36]
			Rv	GGAACATCATTTCTGGACTTTG	GPC	[37]
Xbarc155	4AL (18.2)	59	Fw	GCGAGTATTGACGTCTTATTTTTGAA	PLH, TW	[36]
			Rv	GCGTCATGAATTCTAACAATGTGCATA		
Xwmc617	4BS (6.0)	59	Fw	CCACTAGGAAGAAGGGGAAACT	TKW	[35,38]
			Rv	ATCTGGATTACTGGCCAACTGT	GNS	[37]
Xgwm513	4BL (38.1)	58	Fw	ATCCGTAGCACCTACTGGTCA	GPC	[40]
			Rv	GGTCTGTTCATGCCACATTG		
Xgwm120	5AS (26.0)	58	Fw	GATCCACCTTCCTCTCTCTC	GPC, GL	[34]
			Rv	GATTATACTGGTGCCGAAAC		
Xgwm371	5BL (58.0)	63	Fw	GACCAAGATATTCAAACTGGCC	TKW, TW	[35,39]
			Rv	AGCTCAGCTTGCTTGGTACC		
Xwmc256	6AL (65.5)	61	Fw	CCAAATCTTCGAACAAGAACCC	DH, GY	[36]
			Rv	ACCGATCGATGGTGTATACTGA		
Xbarc178	6BL (50.0)	59	Fw	GCGTATTAGCAAAACAGAAGTGAG	TKW	[35]
			Rv	GCGACTAGTACGAACACCACAAAA		
Xgwm46	7BS (73.9)	60	Fw	GCACGTGAATGGATTGGAC	GPC	[39]
			Rv	TGACCCAATAGTGGTGGTCA	GY	[36]

Chr is chromosome; Ta is annealing temperature; S is short chromosome arm; L is long chromosome arm; Fw is forward primer; Rv is reverse primer; TKW is thousand-kernel weight; TW is test weight; GY is grain yield; YP is yellow pigment content; DH is days to heading; PLH is plant height; GPC is grain protein content; GNS is grain number per spike; GL is gluten content.

**Table 2 genes-14-01155-t002:** Summary of Shannon diversity index (H′) estimated for 104 Ethiopian durum wheat genotypes across all genotypes and for populations, regions, and altitudinal classes.

Grouping Methods			No.	DH	DM	SPS	PLH	TKW	GY	GL	GPC	SPD	VTR	Mean
All genotypes as a single group			104	0.96	0.42	0.84	0.77	0.84	0.8	0.83	0.87	0.93	0.54	0.78
Population	1	West Gojam	8	0.91	0.41	0.87	0.82	0.71	0.84	0.77	0.91	0.82	0.00	**0.71**
	2	East Gojam	8	0.89	0.21	0.89	0.56	0.68	0.83	0.83	0.80	0.97	0.67	**0.73**
	3	North Gondar	8	0.92	0.44	0.70	0.63	0.63	0.75	0.67	0.73	0.94	0.64	**0.71**
	4	South Wello	7	0.80	0.48	0.84	0.58	0.71	0.67	0.86	0.77	0.54	0.55	**0.68**
	5	North Shewa	9	0.98	0.57	0.76	0.71	0.96	0.58	0.85	0.90	0.70	0.55	**0.76**
	6	East Shewa	10	0.92	0.28	0.79	0.76	0.64	0.76	0.68	0.80	0.76	0.57	**0.70**
	7	West Shewa	8	0.97	0.54	0.82	0.75	0.59	0.43	0.97	0.94	0.77	0.54	**0.73**
	8	East Harerge	8	0.94	0.31	0.98	0.76	0.83	0.79	0.73	0.82	0.84	0.00	**0.70**
	9	Bale	7	0.97	0.23	0.65	0.83	0.63	0.88	0.70	0.76	0.96	0.00	**0.66**
	10	Arsi	7	0.74	0.37	0.78	0.69	0.75	0.78	0.69	0.82	0.81	0.37	**0.68**
	11	Central Tigray	7	0.97	0.51	0.92	0.59	0.61	0.58	0.60	0.87	0.77	0.80	**0.72**
	12	Southern Tigray	7	0.65	0.51	0.58	0.83	0.76	0.84	0.69	0.63	0.24	0.73	**0.65**
	13	Cultivars	10	0.78	0.30	0.90	0.64	0.79	0.74	0.84	0.92	0.82	0.38	**0.71**
	Mean			0.89	0.40	0.81	0.71	0.72	0.73	0.77	0.82	0.78	0.45	0.71
Region	1	Amhara	34	0.93	0.38	0.78	0.74	0.67	0.80	0.75	0.80	0.83	0.53	**0.72**
	2	Oromia	46	0.94	0.41	0.90	0.75	0.84	0.77	0.89	0.92	0.90	0.46	**0.78**
	3	Tigray	14	0.89	0.45	0.87	0.65	0.69	0.72	0.69	0.85	0.79	0.64	**0.72**
	Mean			0.89	0.39	0.86	0.70	0.75	0.76	0.79	0.87	0.84	0.5	0.74
Altitudinal class (m.a.s.l)	1	<2000	10	0.89	0.49	0.94	0.75	0.81	0.69	0.86	0.91	0.83	0.38	**0.76**
	2	2001–2400	42	0.95	0.39	0.80	0.67	0.75	0.78	0.83	0.88	0.84	0.64	**0.75**
	3	2401–2800	31	0.91	0.38	0.88	0.75	0.76	0.81	0.81	0.82	0.85	0.36	**0.73**
	4	>2800	11	0.94	0.46	0.86	0.79	0.72	0.73	0.83	0.95	0.91	0.43	**0.76**
	Mean			0.92	0.43	0.87	0.74	0.76	0.75	0.83	0.89	0.86	0.45	0.75

DH is days to heading; DM is days to maturity; SPS is number of spikelet per spike; PLH is plant height; TKW is thousand-kernel weight; GY is grain yield; GC is gluten content and GPC is grain protein content.

**Table 3 genes-14-01155-t003:** Diversity indices of 14 simple sequence repeat (SSR) loci across 13 Ethiopian durum wheat populations.

SSR	Expected Size (bp)	Range of Fragments (bp)	Na	MAF	h	I	Ho	He	Fst	Fis	Fit	Nm	PIC
Xwmc24	122	116–152	4.00	0.70	0.48	0.68	0.36	0.39	0.17	0.06	0.23	1.19	0.45
Xbarc240	267	247–292	6.00	0.40	0.73	1.08	0.08	0.61	0.15	0.87	0.89	1.37	0.69
Xgwm294	102	88–118	5.00	0.37	0.74	1.06	0.25	0.60	0.19	0.58	0.65	1.08	0.70
Xgwm47.1	166	128–178	5.00	0.70	0.48	0.64	0.13	0.36	0.21	0.63	0.71	0.94	0.45
Xbarc12	200	212–257	5.00	0.61	0.57	0.85	0.25	0.49	0.14	0.49	0.56	1.50	0.53
Xgwm493	179	145–179	4.00	0.55	0.58	0.86	0.66	0.53	0.08	−0.24	−0.14	3.03	0.51
Xbarc155	182	180–204	4.00	0.65	0.52	0.61	0.02	0.37	0.26	0.95	0.97	0.73	0.48
Xwmc617	199	196–220	4.00	0.60	0.56	0.72	0.05	0.44	0.20	0.88	0.90	1.01	0.51
Xgwm513	152	122–144	2.00	0.65	0.45	0.63	0.69	0.44	0.04	−0.59	−0.53	6.36	0.35
Xgwm120	162	128–153	3.00	0.61	0.56	0.73	0.17	0.44	0.19	0.62	0.69	1.10	0.50
Xgwm371	191	126–179	5.00	0.49	0.69	0.95	0.45	0.54	0.21	0.17	0.34	0.95	0.65
Xwmc256	117	111–117	2.00	0.89	0.19	0.27	0.10	0.17	0.12	0.37	0.44	1.91	0.17
Xbarc178	266	274–296	3.00	0.55	0.56	0.77	0.00	0.48	0.14	1.00	1.00	1.54	0.48
Xgwm46	187	152–179	4.00	0.38	0.68	0.89	0.02	0.54	0.20	0.97	0.97	0.99	0.62
Mean			4.00	0.58	0.56	0.77	0.23	0.46	0.16	0.48	0.55	1.69	0.50

Na = Number of alleles; MAF = major allelic frequency; h = gene diversity; I = Shannon diversity index; Ho = observed heterozygosity; He = expected heterozygosity; Fst = fixation index; Fis = inbreeding coefficient of an individual relative to the subpopulation; Fit = inbreeding coefficient of an individual relative to the total population; Nm = gene flow; and PIC = polymorphic information content.

**Table 4 genes-14-01155-t004:** Mean genetic diversity indices of 13 populations over 14 SSR loci.

Popuation	N	Na	Ne	I	Ho	He	F	PPL (%)
West Gojam	8.00	2.29	1.87	0.65	0.21	0.42	0.48	93
East Gojam	8.00	2.71	2.14	0.79	0.21	0.48	0.52	93
North Gondar	8.00	2.36	1.80	0.65	0.24	0.41	0.39	100
South Wello	7.00	2.57	1.86	0.68	0.27	0.41	0.32	93
North Shewa	9.00	2.57	1.97	0.72	0.25	0.44	0.42	93
East Shewa	10.00	3.29	2.39	0.95	0.22	0.55	0.58	100
West Shewa	8.00	2.86	2.02	0.79	0.21	0.46	0.56	100
East Harerge	8.00	2.71	1.95	0.73	0.15	0.43	0.65	93
Bale	7.00	2.57	1.86	0.68	0.20	0.40	0.43	93
Arsi	7.00	2.86	2.28	0.87	0.27	0.52	0.42	100
Central Tigray	7.00	2.36	1.79	0.62	0.17	0.38	0.51	93
Southern Tigray	7.00	2.79	2.18	0.81	0.23	0.48	0.51	86
Cultivars	10.00	3.57	2.68	1.04	0.37	0.58	0.36	93
Mean	8.00	2.73	2.06	0.77	0.23	0.46	0.47	95
± SE	0.08	0.07	0.05	0.03	0.02	0.01	0.04	1

N = Sample size; Na = number of alleles per locus; Ne = effective number of alleles; I = Shannon diversity index; Ho = observed heterozygosity; He = expected heterozygosity; F = fixation index; and PPL = percentage of polymorphic loci.

**Table 5 genes-14-01155-t005:** Analysis of molecular variance (AMOVA) showing the partition of within- and among-populations genetic variation of Ethiopian durum wheat germplasm.

Grouping Units	Source	df	SS	MS	Estimated Variance	Percent Variance	Fst	*p*	Nm
Populations	Among	12	276.62	23.05	1.51	12%	0.12	0.001	1.83
	Within	91	1002.97	11.02	11.02	88%			
	Total	103	1279.59		12.53	100%			
Regions	Among	2	43.06	21.53	0.35	3%	0.03	0.003	8.37
	Within	91	1057.39	11.62	11.62	97%			
	Total	93	1100.45		11.97	100%			
Altitudes	Among	3	58.29	19.43	0.38	3%	0.03	0.008	7.70
	Within	90	1042.16	11.58	11.58	97%			
	Total	93	1100.45		11.96	100%			

Df is degrees of freedom; SS is sum of squares; MS is mean squares; Fst is population differentiation; Nm is gene flow.

## Data Availability

Not applicable.

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
