# Peer review of "Genetic Diversity of Durum Wheat (Triticum turgidum L. ssp. durum, Desf) Germplasm as Revealed by Morphological and SSR Markers"

_genes, 2023, doi:10.3390/genes14061155_

Round 1

Reviewer 1 Report

Top of Form

In thIIn current manuscript authors have evaluated the extent and pattern of genetic diversity in the landraces of Durum wheat germplasm from Ethiopian region. For which authors selected diverse landraces and used 14 SSR makers; the results showed a high mean Shannon diversity index (H’ = 0.78) among the genotypes along with a high level of phenotypic variation. Accordingly, the authors put the genotypes into three groups. Both the phenotypic and the molecular analysis produced strong but distinct grouping of cultivars and landraces.

In summary, the report describes a valuable genetic resource and its information for durum wheat; which could be exploited for genetic improvement of wheat meet challenges of global food security in times of war and global warming.

The manuscript could be considered for publication after careful revision.Bottom of Form

Some comments:

1.       Figure captions: please describe the captions in details. Explain the content of figure in its captions.

2.       Line 158, 204, 210, 216, and so on: Reference is missing; Error! Reference source not found.

3.       Table 1. could be included as supplementary information.

4.       For figure 2. Dark colors could be used to indicate the numbers in biplot analysis (PCA).

5.       Line 300 to 309: could be rephrased/written as a paragraph.

6.       Please include key statistical information in conclusions.

7.       Finally, the scientific literature needs to be updated, for instance the introduction and discussion can be enriched with inclusion of additional studies on agricultural crops.

8.       I recommend authors use some other studies on DNA-based molecular markers to improve the scientific literature. See below few examples from literature on molecular marker-based analysis (using RAPD, RFLP, AFLP, SSR, ISSR, ITS, etc): Multiplex molecular marker-assisted analysis of significant pathogens of cotton (Gossypium sp.), 2022; Biocatalysis and Agricultural Biotechnology https://doi.org/10.1016/j.bcab.2022.102557 (Cotton);   Microsatellite and RAPD analysis of grape (Vitis spp.) accessions and identification of duplicates/misnomers in germplasm collection, Upadhyay et al., 2010 Indian J Hortic Volume 67 Pages 8-15; Microsatellite analysis to differentiate clones of Thompson seedless grapevine, Upadhyay et al., 2010, Ind Journal of Horticulture, Volume 67 Issue 2 Pages 260-263

English language is okay.

Assessment of genetic diversity and volatile content of commercially grown banana (Musa spp.) cultivars, Hinge et al., Scientific Reports, 2022; https://doi.org/10.1038/s41598-022-11992-1 (Banana);

Reviewer 2 Report

Comments

Abstract is well written. Add 2-3 lines of result part in the abstract section and how this research is useful.

Kindly include some latest references in the introduction section of the manuscript.

Add one more paragraph of introduction section and write about the Importance of the study and future aspect.

Line no 103 (Error! Reference source not found.) pl. remove

Line no 106-107 degree symbol should be upper case.?

Line no 110 provide the details legends.

Line no 122 describe the details of Genomic DNA extraction

Line no 158-159 (Er- ror! Reference source not found. pl. remove

Line no 204, 210, 214 216, 227, 231, 234, and 239 (Error! Reference source not found.) pl. remove

In Figure 2 improve the image quality and describe the legends.

Line no 282 (Error! Reference source not found.) pl. remove

Line no 278-292 Make it in a single paragraph

Line no 300-310 Make it in a single paragraph

Line no 332-333 provide the details legends.

Line no 364-365 provide the details legends.

In Figure 5 improve the image quality

Line no 409-419 Make it in a single paragraph

Line no 534-535 merged the sentence

Line no 543-544 merged the sentence

Kindly check the grammatical mistake throughout the manuscript.

Minor editing of English language required
